# Highlighting the Biotechnological Potential of Deep Oceanic Crust Fungi through the Prism of Their Antimicrobial Activity

**DOI:** 10.3390/md19080411

**Published:** 2021-07-24

**Authors:** Maxence Quemener, Marie Dayras, Nicolas Frotté, Stella Debaets, Christophe Le Meur, Georges Barbier, Virginia Edgcomb, Mohamed Mehiri, Gaëtan Burgaud

**Affiliations:** 1Laboratoire Universitaire de Biodiversité et Écologie Microbienne, Université de Brest, F-29280 Plouzané, France; quemener.max@gmail.com (M.Q.); nicokayak29@gmail.com (N.F.); Stella.Debaets@univ-brest.fr (S.D.); Christophe.Lemeur@univ-brest.fr (C.L.M.); georges.barbier@univ-brest.fr (G.B.); 2Marine Natural Products Team, Institut de Chimie de Nice, UMR 7272, Université Côte d’Azur, CNRS, 06108 Nice, France; Marie.DAYRAS@univ-cotedazur.fr (M.D.); Mohamed.Mehiri@unice.fr (M.M.); 3Departments of Geology and Geophysics and Biology, Woods Hole Oceanographic Institution, Woods Hole, MA 02543, USA; vedgcomb@whoi.edu

**Keywords:** oceanic crust, fungi, secondary metabolites, molecular screening, antimicrobial assays

## Abstract

Among the different tools to address the antibiotic resistance crisis, bioprospecting in complex uncharted habitats to detect novel microorganisms putatively producing original antimicrobial compounds can definitely increase the current therapeutic arsenal of antibiotics. Fungi from numerous habitats have been widely screened for their ability to express specific biosynthetic gene clusters (BGCs) involved in the synthesis of antimicrobial compounds. Here, a collection of unique 75 deep oceanic crust fungi was screened to evaluate their biotechnological potential through the prism of their antimicrobial activity using a polyphasic approach. After a first genetic screening to detect specific BGCs, a second step consisted of an antimicrobial screening that tested the most promising isolates against 11 microbial targets. Here, 12 fungal isolates showed at least one antibacterial and/or antifungal activity (static or lytic) against human pathogens. This analysis also revealed that *Staphylococcus* *aureus* ATCC 25923 and *Enterococcus* *faecalis* CIP A 186 were the most impacted, followed by *Pseudomonas* *aeruginosa* ATCC 27853. A specific focus on three fungal isolates allowed us to detect interesting activity of crude extracts against multidrug-resistant *Staphylococcus aureus*. Finally, complementary mass spectrometry (MS)-based molecular networking analyses were performed to putatively assign the fungal metabolites and raise hypotheses to link them to the observed antimicrobial activities.

## 1. Introduction

Terrestrial, freshwater, and many marine habitats have been deeply investigated not only regarding fungal diversity and the ecological features of their fungal communities, but also for their ability to produce a large chemodiversity of secondary metabolites recognized as antimicrobials, anticancer agents, immunosuppressive, etc. [1,2]. Marine fungi have been detected in every habitat explored to date, from coastal waters to the deep sea, including hydrothermal vents [3,4], deep subseafloor sediments [5,6,7,8,9] and even the uncharted deep oceanic crust [10,11]. These deep subsurface biosphere fungal communities appear to be mostly represented by ubiquitous taxa, for example, representatives of the genera *Penicillium*, *Aspergillus* or *Cladosporium* [8,11,12], yet these are still active, performing non-trivial functions and ecological roles as deciphered using metatranscriptomic-based approaches. Indeed, using this mRNA-based strategy, fungi were shown to recycle complex organic matter but also to interact with other communities through the expression of biosynthetic gene clusters to produce antimicrobial compounds [9,11,13]. From an ecological angle, these studies highlight important contributions to the complexity and dynamics of the deep biosphere microbiome. The ability of these fungi to produce secondary metabolites with antimicrobial activity in these habitats can be leveraged for use in biotechnological applications.

Fungal cells can be viewed as microbial factories able to synthesize a large array of valuable compounds [14] with one of the most recognized categories to date being secondary metabolites with human health and industrial applications. Studies of bioactive compounds from marine fungi have revealed thousands of novel chemical structures. Interest in these has increased, and there have been ~50 compounds described each year in the early 2000s, and ~300–400 compounds per year since 2013 (Rédou et al., 2016; Blunt et al., 2018; Daletos et al., 2018). While sources of marine fungi that produce valuable natural products are diverse, including fungi associated with algae, sponges, plant- and wood-based habitats, fishes, corals and mollusks [15], increased investigations into marine fungal secondary metabolites are also explained by an increasing interest in deep-sea fungi [16].

Antibiotics have been overconsumed and misused in recent years and issues related to antibiotic resistance have emerged, jeopardizing human health [17]. Increases in either incidence and lethality of infectious diseases caused by pathogenic and multidrug-resistant bacteria require a deep renewal of the antibiotic arsenal, and this effort is now supported by numerous initiatives to address antibiotic resistance solutions [18,19]. Exploration of marine fungal culture collections and new isolates from atypical uncharted habitats, such as the deep biosphere, could lead to the discovery and isolation of novel antimicrobial compounds. Indeed, transcript fungal data reported using metatranscriptomic-based approaches include organic matter recycling [13] but also biosynthesis of antimicrobial compounds, suggesting possible adaptive capabilities for competing with other microorganisms [9,11]. The aim of this study was thus to confirm these conclusions by conducting a first screening of the biotechnological potential of a previously established fungal culture collection from the deep oceanic crust [11], representing one of the last great frontiers for biological exploration on Earth, with a specific focus on their ability to produce antimicrobial compounds. The biotechnological potential of these fungal isolates was assessed using a two-step bioprospecting approach to first identify their genetic potential, i.e., the presence of specific biosynthetic gene clusters, and then to assess their ability to produce antimicrobial compounds that are effective against 11 common human pathogens using an agar plug diffusion assay. Following this genetic mining and antimicrobial screening approach, complementary mass spectrometry (MS)-based molecular networking analyses were performed to annotate the putative metabolites and explain the observed antimicrobial activities.

## 2. Results and Discussion

### 2.1. Presence of Genes Involved in the Production of Secondary Metabolites

The first screening was processed on 75 fungal isolates which were extracted from the whole collection of 128 deep oceanic crust fungi based on their taxonomy. Non-ribosomal peptide synthetase (NRPS) genes were observed in 79.7% of the isolates, followed by type III polyketide synthases (PKS), terpene synthase (TPS) and type I PKS genes detected in 47.3%, 44.6% and 40.5% of the isolates, respectively (Figure 1). Among the 58 filamentous fungi, 57 harbored at least one of the targeted genes. Indeed, except for one isolate affiliated to *Bjerkandera adusta* (UBOCC-A-118179), 13 isolates display one gene, 14 isolates have two different genes, 15 isolates have three different genes and 15 isolates possess four different genes, each of these conditions representing ~1/4th of the filamentous fungal collection (Figure 1A). Among the 17 yeasts, only five harbored at least one of the targeted genes: one isolate (6%) has one gene, two isolates (12%) have three different genes and two isolates (12%) display four different genes (Figure 1B). As a special note, the presence of representatives of the genus *Entyloma* in phylogenetic trees of both filamentous fungi and yeasts is explained by different observed morphological features. The isolate identified as *Entyloma calendulae* (UBOCC-A-118153) grew as a thallus and was thus considered as a filamentous fungus, while the isolates identified as *Entyloma ficariae* (UBOCC-A-218010, UBOCC-A-218013 and UBOCC-A-218014) exhibited colonies on Petri dishes characteristic of yeasts.

Among the most represented genera, i.e., *Penicillium* and *Aspergillus*, every isolate displayed at least one targeted gene, with a median gene representation of two genes for *Penicillium* and three genes for *Aspergillus*. Isolates affiliated to the genus *Penicillium* show a high occurrence of NRPS (retrieved in ~95% of the isolates) and close distribution for type III PKS, TPS and type I PKS with an occurrence of 45%, 45% and 38%, respectively. Regarding isolates affiliated to the genus *Aspergillus*, the genetic potential was dominated by NRPS (retrieved for all six isolates) and followed by type III PKS and TPS (representing ~85% of the *Aspergillus* isolates) and type I PKS (~65%). Isolates affiliated to the genera *Cladosporium, Sistotrema* and *Microscypha* harbor an important genetic potential with a median of 4, 3.5 and 3 represented genes, respectively. Finally, two genera represented by a single isolate also display an important genetic potential, i.e., *Neobulgaria* sp. (UBOCC-A-118154) and *Ramularia coryli* (UBOCC-A-118163).

Representatives of the genera *Aspergillus, Penicillium* and *Cladosporium* are frequently retrieved in marine habitats, from coastal waters to the deep biosphere, for example, hydrothermal vents and deep subseafloor sediments [8,11,21]. Such taxa appear to be of great interest in terms of biotechnological potential thanks to their abilities to produce a diversity of secondary metabolites, including antimicrobial compounds [22,23,24,25]. Compared with *Aspergillus*, *Penicillium* and *Cladosporium,* marine members of the genera *Microscypha* and *Sistotrema* appear to be more limited in their diversity of secondary metabolites produced [26]. In general, filamentous fungi appear to have greater potential to produce secondary metabolites than yeasts as evidenced here by the presence of a relatively high occurrence of specific biosynthetic gene clusters. This is consistent with a previous study based on deep subsurface fungi [8]. Surprisingly, none of the filamentous fungal isolates in this study seemed to possess PKS/NRPS hybrid genes, not even *Aspergillus* and *Penicillium* isolates, for which hybrid PKS/NRPS were detected in previous studies [8,27].

No specific distribution patterns of BGCs were visualized, neither correlating with genus or species nor depending on the isolation depth (linear regression, *p*-values > 0.05), thus illustrating high intraspecific variability.

### 2.2. Antimicrobial Activities

One of the aims of our biotechnological screening was to select the most promising isolate candidates for bioactivity screening by examining our collection through the prism of their genetic potential. Here, 23 fungal isolates (13 filamentous fungi and 10 yeasts) were selected on this basis but also on the basis of their growth rate so that we avoided slow-growing isolates. We recognized that this strategy may lead us to missing interesting isolates but the idea here was to lower the number of tests for our one strain many compounds (OSMAC)-like approach. In total, from the 1518 tests processed (23 isolates grown on six media and tested against 11 microbial targets), 99 positive microbial activities were observed which represented 6.5% of tested conditions. Among these 99 antimicrobial activities, 83 were related to bactericidal/fungicidal activities (84%) and 16 to bacteriostatic/fungistatic ones (16%). Out of the 83 bactericidal/fungicidal activities, 71 were against Gram-positive bacteria (70%), 25 against Gram-negative bacteria (25%) and 6 against the yeast *Candida albicans* (5%).

None of the yeasts tested (10 isolates) displayed an antimicrobial activity (data not shown), which appears to be consistent with previous studies comparing yeasts to filamentous fungi in terms of secondary metabolites [8,28]. This absence of antimicrobial activity may reflect their lower observed genetic potential to produce secondary metabolites based on the gene targets we selected, which was lower compared to the filamentous fungi we examined. However, these observations may not extend to all yeasts or filamentous fungi. As an example, the filamentous *Leptosphaeria dryadis* (UBOCC-A-118058) showed a high genetic potential by having three of our targeted genes (type III PKS, NRPS and TPS), yet it did not show any antimicrobial activities. This suggests that the gene targets we focused on may play other roles in the growth of that fungus, and could possibly have antiviral, immunosuppressive, anti-inflammatory, antioxidant or antitumor activities. They may also be involved in resistance to environmental biotic or abiotic stresses, as previously shown [28,29,30,31,32]. In contrast, *Penicillium olsonii* (*P. olsonii*) (UBOCC-118169), which presented a low genetic potential with two represented genes, showed high antimicrobial activities against five pathogens, namely *Escherichia coli* (*E. coli*) ATCC 25922, *Salmonella enterica* (*S. enterica*) CIP 8297, *Pseudomonas aeruginosa* (*P. aeruginosa*) ATCC 27853, *Enterococcus faecalis* (*E. faecalis*) CIP A 186 and *Staphylococcus aureus* (*S. aureus*) ATCC 25923 (Figure 2). This taxon may potentially have additional genes for other secondary metabolites that were not targeted in our study. Another possible explanation for some contrasting results is that our designed primers likely do not target an exhaustive list of genes of interest. This molecular screening can thus be viewed as providing potentially incomplete information on the potential for secondary metabolite activity and appears as a filtering tool to lower the number of isolates to be screened but which is not exempt from biases. A more complete, but admittedly more expensive and time-consuming, approach would be genome sequencing of all isolates followed by gene analysis targeting synthesis of secondary metabolites through the AntiSMASH tool [32,33,34,35]. This would produce a more complete picture of biosynthetic gene clusters in each organism and would allow for chemical structure prediction and networking approaches [36]. This would be the logical next step after conducting a large-scale screening of a culture collection as done in this study. Finally, another explanation to better understand these contrasting results is that our approach did not integrate the dynamic nature of the fungal metabolome. Indeed, antimicrobial activity was screened at one specific time frame (3 days for yeasts, and 7 to 14 days for filamentous fungi). However, as demonstrated in a previous study based on time-scale metabolomics [37], only subsets of BGCs are expressed at a given time, suggesting that only a fraction of the metabolome was detected here.

The effect of the culture medium used for growth was also tested here. Based on the 99 positive hits for antimicrobial activities, 29, 24, 13, 12, 11 and 10 positive hits were obtained from MEPA, PDA, Czapek, Wickerham-Chitin, Wickerham-Halogen and Wickerham medium, respectively. This clearly highlights the importance of diversification of culture media for this kind of screening and appears to be consistent with many metabolomic studies that show the utility of the OSMAC approach for stimulating the marine fungal metabolome [38,39,40,41]. Here, the MEPA medium, enriched with lignin, chitin and cellulose, produced a relatively large number of positive hits. The presence of such complex polymers seems to enhance the synthesis of antimicrobial compounds, which has, to the best of our knowledge, not been previously reported.

Finally, 52% of the tested fungal isolates produced antimicrobial compounds. This is consistent with a previous study that used a similar strategy, and highlighted that 33% of a collection from deep subsurface sediments produces antimicrobial compounds [28], but also with another one focusing on the antibacterial activity of marine fungi from the Arctic, highlighting 50% of antibacterial activity against human pathogens [42]. Our results are also in agreement with previous bioprospection screening reports that ~50% of microbial isolates from the black coral *Antipathes dichotoma* exhibited antimicrobial activities [43] but contrast with some reports showing ~10% activity for deep sediment bacteria [44] or for microbial isolates obtained from sea anemone and holothurian samples [45]. In summary, the deep biosphere and its associated fungal diversity is a promising reservoir of putatively interesting biomolecules.

### 2.3. Antimicrobial Activity against Multidrug-Resistant Bacteria

Based on the antimicrobial activity spectrum that we observed, it appears that two isolates affiliated to *P. olsonii* (UBOCC-A-118169 and UBOCC-A-119011) and one isolate affiliated to *Aspergillus conicus* (*A. conicus*) (UBOCC-A-118156) display the most interesting antimicrobial profiles. Indeed, the isolates affiliated to *P. olsonii* were active against a wide range of pathogens and the isolate affiliated to *A. conicus* was highly active against *S. aureus*, regardless of the medium that was used to produce biomass (Figure 2). As the Gram-negative bacterium *P. aeruginosa* and the Gram-positive bacterium *S. aureus* targets were highly impacted by these isolates with obvious bactericidal activities, we decided to conduct a second antimicrobial screening using multidrug-resistant *P. aeruginosa* and *S. aureus*. The idea here was to generate biomass for these three isolates and to extract secondary metabolites using different combinations of solvents in order to obtain crude extracts to test for associated antimicrobial activities.

Preliminary results on crude extracts allowed us to confirm the antimicrobial activity of *P. olsonii* (UBOCC-A-118169) against *S. aureus*, and also revealed activity against *S. aureus* MecA III. The obtained CMI were between 8–16 µg/mL and 4–8 µg/mL for *S*. *aureus* and *S. aureus* MecA III, respectively. No activities were obtained from the crude extracts obtained from the other strains, suggesting that the compound(s) responsible for the detected antimicrobial activity was (were) not extracted using our methods.

### 2.4. Molecular Networking

Metabolomic fingerprints, associated with each organic extract (F1 and F2, see M&M), were obtained by (i) HPLC-PDA-ELSD and (ii) UHPLC-HRESIMS (/MS), allowing us to characterize the chemical diversity of the extracts, making the evaluation of the number, the relative amounts, the chemical family and the molecular mass of the metabolites constituting the extracts possible. HPLC-DAD-ELSD analyses of the organic extract F1 (EtOAc/CH_2_Cl_2_ extracts, see M&M) from the three strains revealed interesting chemical fingerprints (Appendix A). The metabolomic profiles of the two *P. olsonii* (UBOCC-A-118169 and UBOCC-A-119011) isolates appeared similar. Regarding *A. conicus* (UBOCC-A-118156), the organic extract exhibited one major metabolite and several eluted compounds late in the series that are predicted to be fatty acids.

The MS/MS spectra for each metabolite were compared to several databases (MarinLit, Dictionary of Natural Products (DNP), Natural Products Atlas (NPA), Scifinder and Chemspider) and to published data [46,47,48,49]. The main compounds produced by the two isolates affiliated to *P. olsonii* (UBOCC-A-118169 and UBOCC-A-119011) were assigned to xanthoepocin and asperphenamate. Using a similar approach, the major compound produced by *A. conicus* (UBOCC-A-118156) was assigned to asperphenamate.

To putatively annotate the fungal metabolites that may be responsible for the detected antimicrobial activities, the organic extracts were analyzed by HPLC-HRMS to build feature-based molecular networks. The graphical representation of the molecular network (depicting the chemical space present in MS/MS experiments) of *P. olsonii* (UBOCC-A-118169 (red), UBOCC-A-119011 (blue)) isolates using MS/MS data of each metabolite constituting the crude organic extracts F1 allowed us to highlight 1447 nodes and 1805 edges which suggest the production of numerous metabolites (Figure 3). Almost all nodes are common to both *P. olsonii* isolates, confirming the similarity of the metabolomic profiles. Among the nodes, a comparison of the experimental MS/MS spectra with the ones available in GNPS Public Spectral Libraries (https://gnps.ucsd.edu/ProteoSAFe/libraries.jsp, accessed date in March 2021) allows us to highlight a putative subcluster of xanthoepocin containing 10 nodes and 12 edges, and a putative subcluster including asperphenamate containing 40 nodes and 105 edges. The presence of different compounds in the same cluster suggests the possibility of a common biosynthetic pathway or some metabolites that could be analogues of xanthoepocin and asperphenamate, respectively. The UBOCC-A-119011 strain (blue) contains many putative xanthoepocin analogues, two of which are not present in the UBOCC-A-118169 strain (red). Previous studies have shown that xanthoepocin inhibits the growth of *S. aureus* and *S. aureus* MRSA with MICs of 1.56 µg/mL [47]. Although asperphenamate is well known for its antitumor activity [50,51], it is inactive against the microorganisms tested (*S. aureus* and MRSA) at up to 64 µg/mL [48]. Combining all these results, but recognizing that our putative MS-based molecular networking annotation has to be taken with caution, *P. olsonii* UBOCC-A-118169 and *P. olsonii* UBOCC-A-119011 may produce xanthoepocin-like molecules as antibacterial secondary metabolites. Complementary analyses to isolate the xanthoepocin analogues produce by these strains would allow access to new molecules with potential antibacterial activity.

Using a similar strategy, the molecular network for the *A. conicus* (UBOCC-A-118156) strain using MS/MS data of the crude organic extract F1 is constituted by 1209 nodes and 1562 edges. Among the constellation of nodes, the putative subcluster of the main compound, assigned by comparing the obtained MS/MS data with that of the pure metabolite, contains 61 nodes and 161 edges (Figure 4). In order to assign analogues, MS/MS spectra were compared to published data [49]. We used the asperphenamate analogue naming system proposed by Subko et al. (2021). Putative assigned analogues mainly differ from asperphenamate by the type of amino acid, for example, asperphenamate Y is substituted by tyrosine instead of phenylalanine. Asperphenidine is an analogue constituted by phenylalanine and nicotinic acid (Figure 5). Recently, some new natural asperphenamate analogues have been isolated from several fungi, such as 4-OMe-asperphenamate [48,52] and asperphenamates B and C [53]. Moreover, 22 new asperphenamate analogues were isolated from *Penicillium astrolabium* and assigned by HRMS/MS. Some derivatives were additionally confirmed by isolation, NMR analyses and structure elucidation [48]. Although the antibacterial activity of these compounds has not been evaluated, asperphenamates Y and W and asperphenidine F1 were evaluated against five cancer cell lines and moderate activities were detected against breast, skin, liver and pancreas cell lines [49]. The large number of nodes in the putative asperphenamate cluster within the *A. conicus* (UBOCC-A-118156) molecular network may indicate that many analogues remain to be isolated and characterized. Such analogues may be linked to antibacterial activity that we detected in *P. conicus* (UBOCC-A-118156) but, as emphasized for the *P. olsonii* MS-based molecular network analyses, these results have to be taken with caution since the metabolites were putatively assigned.

## 3. Materials and Methods

### 3.1. Culture Collection

The analyzed culture collection comprises fungi isolated from deep lower oceanic crust samples from below Atlantis Bank, the Indian Ocean, from 10 to 780 m below the seafloor (mbsf) during the International Ocean Discovery Program Expedition 360 [10]. Several genetic markers were used to identify 128 fungal isolates, and isolates were extensively characterized in terms of their ecophysiological and metabolic features to assess their environmental relevance [11]. All isolates are preserved in the UBO Culture Collection (https://www.univ-brest.fr/ubocc, accessed date in July 2021) and are available upon request.

### 3.2. Occurrence of Genes Involved in Secondary Metabolite Anabolism

A selection of 75 fungal isolates from among the 128 isolates in the culture collection was made based on taxonomy in order to cover the broadest extent of the culturable diversity. Extracted DNA from each isolate was used as a template to amplify secondary metabolite genes of interest, including types I and III polyketide synthase (PKS), non-ribosomal peptide synthetase (NRPS), hybrid PKS/NRPS and terpene synthase (TPS) genes, using degenerated primers and PCR conditions detailed in Rédou et al. (2015). A value between 0 and 3 was assigned to represent the number of genes detected in each isolate, as a proxy for their molecular potential; 0 meaning no detected genes, 1 meaning 1 gene, 2 meaning 2 genes and 3 meaning more than 2 genes.

### 3.3. First Screening—Agar Plug Diffusion Assay

Antimicrobial screening was conducted using six different media with a fixed concentration of 3% sea salts. We used potato dextrose agar (PDA), Czapeck Dox medium (CZ), Wickerham medium (W), MEPA supplemented with chitin and cellulose at 1 g/L (MP), Wickerham supplemented with chitin at 5 g/L (WC), and Wickerham supplemented with the halogens potassium bromide at 0.09 g/L, potassium chloride at 0.6 g/L and sodium fluoride at 0.02 g/L (WH). Fungal isolates were inoculated on each medium at 20 °C and grown for 3 days for yeasts, and 7 to 14 days for filamentous fungi, depending on their growth rate. At the same time, 11 human microbial pathogens including Gram-negative, Gram-positive and a yeast were precultured on tryptone salt broth (TSB) for 24 h at 30 °C or 37 °C depending on the pathogen. Specific agar media were inoculated with each pathogen (Table 1) at a standardized concentration of 10^6^ cells/mL. For the diffusion assay, 12 agar plugs were removed from a selection of 23 fungal isolates at evenly spaced locations on each yeast or bacterial pathogen plate and the empty holes were replaced by agar plugs containing actively growing mycelium or yeast cells removed from the 6 different culture media. Each combination of pathogen and fungal isolate was tested in triplicate and negative controls were included without fungal inoculation (agar plugs from sterile media). Plates were then incubated at 30 °C or 37 °C for 72 h and observed every 24 h to differentiate bacteriostatic/fungistatic from bactericidal/fungicidal activities.

### 3.4. Secondary Screening—Metabolite Extraction, Molecular Networking and Determination of the Minimum Inhibitory Concentration

The most promising fungal isolates, based on the agar plug screening, were selected for downstream analyses, including extraction of secondary metabolites and antimicrobial assays on multidrug-resistant (MDR) pathogens. Around 80 Petri dishes were used to produce sufficient fungal biomass for each selected isolate. Dried biomass was suspended in 50 mL of a mixture of EtOAc:CH_2_Cl_2_ (1:1, *v*/*v*), homogenized with an Ultra Turrax (IKA, USA) device, extracted three times by sonication with 50 mL of EtOAc:CH_2_Cl_2_ (1:1, *v*/*v*) and filtered. The combined filtrates were evaporated under reduced pressure to yield a first crude organic extract (F1). The biomass was then suspended in 50 mL of a mixture of MeOH:CH_2_Cl_2_ (1:1, *v*/*v*), extracted three times by sonication with 50 mL of MeOH:CH_2_Cl_2_ (1:1, *v*/*v*) and filtered. The combined filtrates were evaporated under reduced pressure to yield a second crude organic extract (F2). Each crude organic extract, F1 and F2, was first re-suspended in a mixture of MeOH:CH_2_Cl_2_ (1:1, *v/v*) to reach a final concentration of 10 mg/mL and then analyzed by HPLC-PDA-ELSD. The latter was performed with a Waters Alliance 2695 high-performance liquid chromatography (HPLC) system (Waters Corporation, Milford, MA, USA) coupled with a Waters 996 photodiode array detector and a Sedex 55 evaporative light-scattering detector (SEDERE, France), using a bifunctional Macherey-Nagel NUCLEODUR^®^ Sphinx RP column (250 × 4.6 mm, 5 μm) consisting of a balanced ratio of propylphenyl and C18 ligands. The mobile phase was composed of H_2_O (plus 0.1% HCO_2_H) and acetonitrile (CH_3_CN plus 0.1% HCO_2_H) and the following gradient was used: H_2_O:CH_3_CN 90:10 for 5 min, H_2_O:CH_3_CN 90:10 to 0:100 for 30 min, 0:100 for 5 min, 0:100 to 90:10 for 15 min (flow: 1.0 mL·min^−1^, injection volume: 20 µL). Chromatograms were extracted at the following detection wavelengths for visual inspection: 214, 254 and 280 nm. Moreover, each crude organic extract was analyzed by HPLC/ESI-MS/MS in both positive and negative ion modes using a Vanquish UHPLC coupled with a Thermo Q-Exactive (UPLC-HRMS) Orbitrap (Thermo Fisher Scientific GmbH, Bremen, Germany) and an ESI source operated with the Xcalibur (Version 2.2, ThermoFisher Scientific) software package. A bifunctional Macherey-Nagel NUCLEODUR^®^ Sphinx RP column (150 × 4.6 mm, 3 μm) consisting of a balanced ratio of propylphenyl and C18 ligands was used with an injection volume of 5 µL and a flow rate of 0.3 mLmin^−1^. The mobile phase was composed of H_2_O (plus 0.1% HCO_2_H) and acetonitrile (CH_3_CN plus 0.1% HCO_2_H) and the following gradient was used: H_2_O:CH_3_CN 90:10 for 5 min, H_2_O:CH_3_CN 90:10 to 0:100 for 30 min, 0:100 for 5 min, 0:100 to 90:10 for 15 min. HR-MS/MS raw data files were converted from RAW to mzXML format using MSConvert software, and clustered with MS-Cluster using Global Natural Products Social Molecular Networking (GNPS) [21]. Molecular networks were created using the online workflow of GNPS. The following settings were used for generation of the *P. olsonii* molecular network: minimum pairs cos 0.7; parent mass tolerance, 0.02 Da; ion tolerance, 0.02; network topK, 6; minimum matched peaks, 7; minimum cluster size, 2. The following settings were used for generation of the *A. conicus* molecular network: minimum pairs cos 0.8; parent mass tolerance, 0.02 Da; ion tolerance, 0.02; network topK, 6; minimum matched peaks, 7; minimum cluster size, 2. Data were visualized and analyzed using Cytoscape 3.8.1 [54].

The obtained dried crude organic extracts were dissolved in pure dimethyl sulfoxide (DMSO) and integrated into TSB medium used here to culture the 8 selected pathogens. The concentration range tested for each crude extract was between 256 µg/mL and 2 µg/mL (7 two-fold dilutions) in a 96-well plate. Suspensions of pathogens were calibrated at 10^6^ cells/mL. The final concentration of DMSO was less than 1% in each condition and did not affect the microbial growth (as visualized in our positive controls). Among the 8 selected pathogens, 4 bacteria were multidrug resistant (Table 2). Plates were incubated at 37 °C for 48 h. To highlight the minimum inhibitory concentration (MIC), bacterial growth was determined using laser nephelometry (BMG Labtech), used here as a mid-throughput method to assess bacterial growth. Finally, the MIC was described as the lowest concentration that resulted in complete inhibition of microbial growth, as described in CLSI standard M07-A9.

## 4. Conclusions

Out of 75 fungal isolates (filamentous fungi and yeasts) selected from a collection of 128 deep oceanic crust fungi, 62 were identified as putatively interesting based on the presence of genes usually involved in the production of secondary metabolites. Based on the pattern of BGCs that each isolate contained, 23 isolates were screened for antimicrobial activities using an OSMAC-like approach. This screening highlighted 12 isolates showing at least one antibacterial and/or antifungal activity (static or lytic) against human pathogens. Approximately ~55% of the 23 targeted isolates showed activity. Among the 11 pathogens tested, the Gram-positive bacterial strains *S. aureus* ATCC 25923 and *E. faecalis* CIP A 186 were the most impacted by the fungal strains tested, followed by the Gram-negative bacterium *P. aeruginosa* ATCC 27853. Three fungal isolates were selected based on their demonstrated antimicrobial potential when tested against a multidrug-resistant *S. aureus*. Out of these three isolates, crude extracts from *P. olsonii* (UBOCC-A-118169) revealed antimicrobial activity against this pathogen. Finally, a metabolomic approach based on molecular networking allowed us to examine the metabolome of these three isolates and highlighted the putative occurrence of numerous compounds for which complementary analyses are needed. Indeed, our strategy here was to highlight the biotechnological potential of a unique and invaluable collection of deep oceanic crust fungal isolates using a polyphasic approach merging genetic mining and antimicrobial screening. The constellations of putative compounds and their analogues visualized using MS-based molecular networking definitely call for in-depth analyses in order to obtain larger amounts of biomass and thus finely characterize each compound. This study demonstrates that fungi from uncharted extreme habitats display untapped biotechnological potential and that the deep sea, and more precisely, the deep biosphere, may harbor a complex diversity of novel compounds waiting to be characterized.

## Figures and Tables

**Figure 1 marinedrugs-19-00411-f001:**
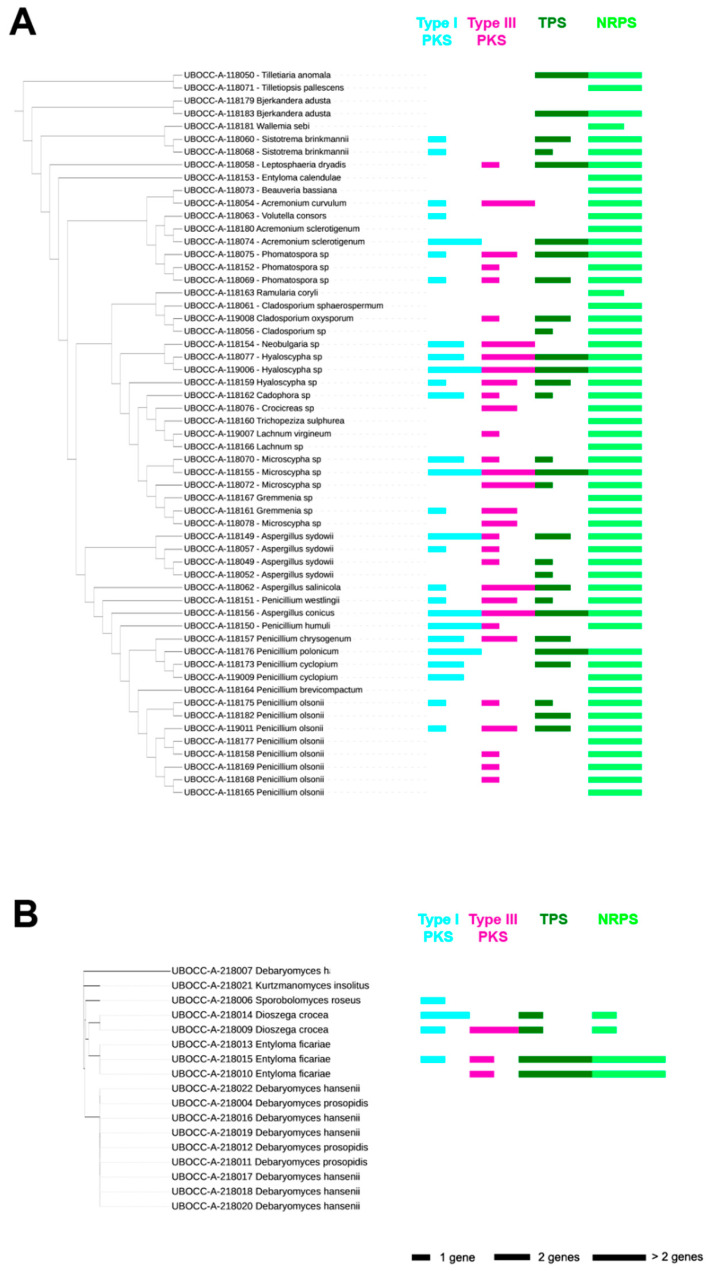
Presence/absence of genes coding type I and III PKS, NRPS, hybrid PKS-NRPS and TPS. Dataset of filamentous fungal ITS fingerprints (**A**) and yeast 26S fingerprints (**B**) coupled with type I (clear blue), type III (pink) PKS, NRPS (clear green), hybrid PKS/NRPS (dark blue) and TPS (dark green) gene occurrence using aligned multivalue bar chart. This figure was generated using Interactive Tree of Life v2 [20].

**Figure 2 marinedrugs-19-00411-f002:**
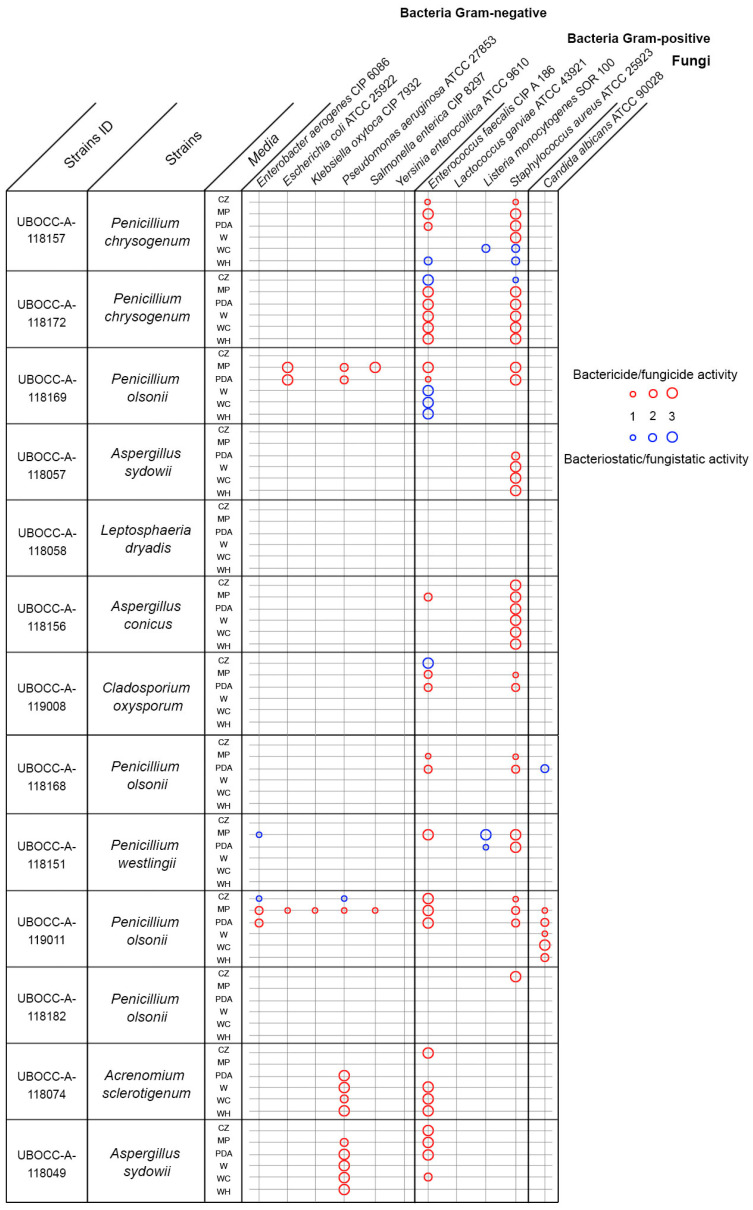
Antimicrobial spectrum of the 14 antibiotic-producing fungi isolated from the oceanic crust. Inhibition rates vary between 0 (no inhibition) and 3 (complete inhibition). Full boxes correspond to lytic antimicrobial activity and crossed boxes to static antimicrobial activity.

**Figure 3 marinedrugs-19-00411-f003:**
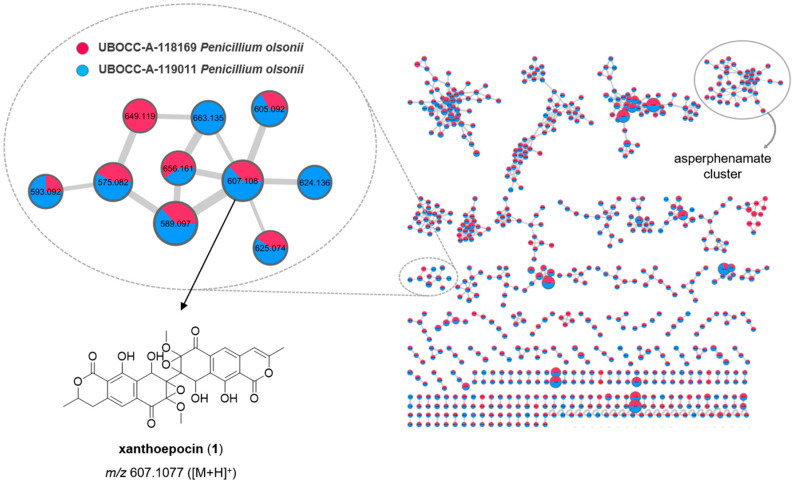
Molecular network constructed using MS/MS data from the crude organic extracts F1 of *P. olsonii* (UBOCC-A-118169 (red) and UBOCC-A-119011 (blue)). The subcluster of one of the main compounds, xanthoepocin, of *P. olsonii* is enlarged.

**Figure 4 marinedrugs-19-00411-f004:**
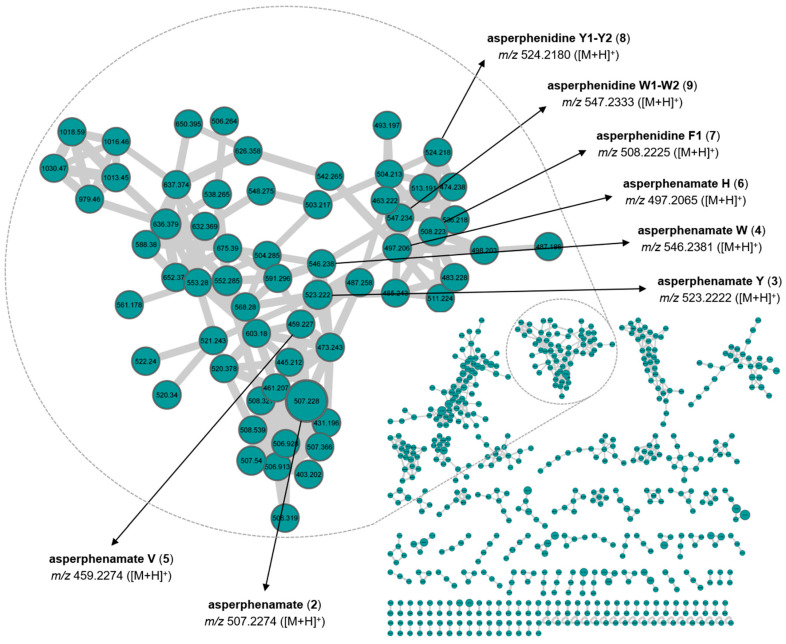
Molecular network constructed using MS/MS data from the crude organic extract F1 of *A. conicus* (UBOCC-A-118156). The subcluster of one of the main compounds, asperphenamate, of *A. conicus* is enlarged.

**Figure 5 marinedrugs-19-00411-f005:**
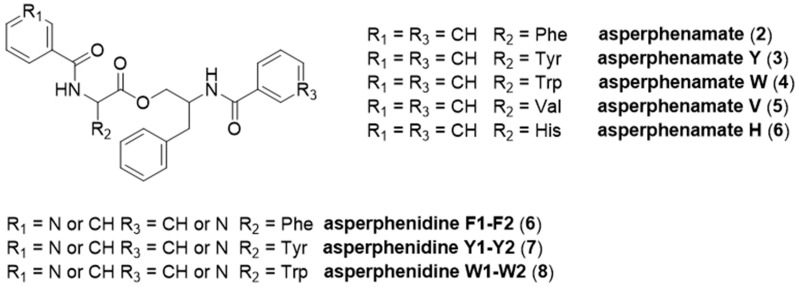
Chemical structures of asperphenamate analogues.

**Table 1 marinedrugs-19-00411-t001:** Temperature conditions and media used for each of the 11 targeted human pathogens.

Microbial Pathogens	Temperature	Medium
*Enterobacter aerogenes* CIP 6086	30 °C	TSB
*Escherichia coli* ATCC 25922	37 °C	TSB
*Klebsiella oxytoca* CIP 7932	37 °C	TSB
*Pseudomonas aeruginosa* ATCC 27853	37 °C	TSB
*Salmonella enterica* CIP 8297	37 °C	TSB
*Yersinia enterocolitica* ATCC 9610	30 °C	TSB + NaCl
*Enterococcus faecalis* CIP A 186	37 °C	TSB
*Lactococcus garviae* ATCC 43921	30 °C	BHI
*Listeria monocytogenes* SOR 100	37 °C	BHI
*Staphylococcus aureus* ATCC 25923	37 °C	TSB
*Candida albicans* ATCC 90028	37 °C	Yeast medium

**Table 2 marinedrugs-19-00411-t002:** Tested human pathogens for the second antimicrobial screening.

Basic Pathogens	MDR Pathogens
*Enterococcus faecalis* CIP A 186	*Pseudomonas aeruginosa* VIM2
*Staphylococcus aureus* ATCC 25923	*Pseudomonas aeruginosa* Brse
*Escherichia coli* ATCC 25922	*Staphylococcus aureus* MecC
*Pseudomonas aeruginosa* ATCC 27853	*Staphylococcus aureus* MecA III

## Data Availability

Not applicable.

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
