# Peer review of "Highlighting the Biotechnological Potential of Deep Oceanic Crust Fungi through the Prism of Their Antimicrobial Activity"

_marinedrugs, 2021, doi:10.3390/md19080411_

Round 1

Reviewer 1 Report

The manuscript describes the biological potential of deep-sea fungi supported by the presence of BGCs and the antimicrobial activity of the extracts. The experiments were designed and carried out appropriately, and the conclusions have been drawn based on solid experimental evidences and unambiguous interpretation of those. The depiction of the results (Tables and Figures) is very neat and understandable, and descriptions are presented for good readability. Overall this manuscript is of high quality and has significance in view of process development. Some minor concerns are as below. - The total number of strains analyzed should be provided before discussing proportions containing BGCs (page 1, line 80). It seems that the analyzed isolates are those descibed in reference 11 and the number is provided in the experimental and conclusion section of this manuscript. However, it would be better if a brief description on the analyzed strains is provided at the beginning of the 'results and discussion'. - Some words (such as Biosynthetic Gene Clusters, Non-Ribosmal Peptide Synthetase, or Polyketide Syntases and etc.) do not have to start with a capital letter when used in sentences.

Author Response

The manuscript describes the biological potential of deep-sea fungi supported by the presence of BGCs and the antimicrobial activity of the extracts. The experiments were designed and carried out appropriately, and the conclusions have been drawn based on solid experimental evidences and unambiguous interpretation of those. The depiction of the results (Tables and Figures) is very neat and understandable, and descriptions are presented for good readability. Overall this manuscript is of high quality and has significance in view of process development. Some minor concerns are as below. - The total number of strains analyzed should be provided before discussing proportions containing BGCs (page 1, line 80). It seems that the analyzed isolates are those described in reference 11 and the number is provided in the experimental and conclusion section of this manuscript. However, it would be better if a brief description on the analyzed strains is provided at the beginning of the 'results and discussion'. - Some words (such as Biosynthetic Gene Clusters, Non-Ribosomal Peptide Synthetase, or Polyketide Synthases and etc.) do not have to start with a capital letter when used in sentences.

Thanks a lot for your positive comments on our manuscript and your overall impression that our ms is of high quality. Below you’ll find answers to your comments:

- « The total number of strains analyzed should be provided before discussing proportions containing BGCs (page 1, line 80). It seems that the analyzed isolates are those described in reference 11 and the number is provided in the experimental and conclusion section of this manuscript. However, it would be better if a brief description on the analyzed strains is provided at the beginning of the 'results and discussion ».

Thank you for this comment. This information is definitely important. Thanks for pointing this out. We have modified the text and provided this information: The first screening was processed on 75 fungal isolates which were extracted from the whole collection of 128 deep oceanic crust fungi based on their taxonomy

-  « Some words (such as Biosynthetic Gene Clusters, Non-Ribosomal Peptide Synthetase, or Polyketide Syntases and etc.) do not have to start with a capital letter when used in sentences »

Thanks for this comment. Modifications have been processed.

Reviewer 2 Report

This work describes the biotechnological potential of deep oceanic crust fungi as resource of antimicrobial agents. It is an interesting study, which is very well done and well written.

In my opinion the paper can be accepted after minor modifications:

  1. Typographical error in the title: Pag. 1, line 2: “Oceanic” instead of “Ooceanic”.
  2. The abstract should be shortened a little. The abstract should be a total of about 200 words maximum.
  3. References should be checked:

         - The name of the microorganism needs to be written in italics (ref. 22, 23, 24, 25, 36, 37, 3941, 43, 46, 47, 49, 52, 53, 54).

         - The name of the journal should be abbreviated in all references.

         - Some references are uncompleted: pages number is missing in references 2, 9, 11, 28, 34, 36, 37, 40, 41, 49, 52, and volume in reference 11.

In summary, this paper includes so useful information but requires minor modifications to make it suitable for publication.

Author Response

This work describes the biotechnological potential of deep oceanic crust fungi as resource of antimicrobial agents. It is an interesting study, which is very well done and well written.

In my opinion the paper can be accepted after minor modifications:

  • Typographical error in the title: Pag. 1, line 2: “Oceanic” instead of “Ooceanic”.
  • The abstract should be shortened a little. The abstract should be a total of about 200 words maximum.
  • References should be checked:

         - The name of the microorganism needs to be written in italics (ref. 22, 23, 24, 25, 36, 37, 3941, 43, 46, 47, 49, 52, 53, 54).

         - The name of the journal should be abbreviated in all references.

         - Some references are uncompleted: pages number is missing in references 2, 9, 11, 28, 34, 36, 37, 40, 41, 49, 52, and volume in reference 11.

In summary, this paper includes so useful information but requires minor modifications to make it suitable for publication.

Thanks a lot for your positive comments on our manuscript and many thanks for emphasizing the fact that our manuscript is interesting, well done and well written. Below you’ll find answers to your comments:

    1      Typographical error in the title: Pag. 1, line 2: “Oceanic” instead of “Ooceanic”.

Thanks. This typo has been corrected.

    2      The abstract should be shortened a little. The abstract should be a total of about 200 words maximum.

Thanks for pointing this out. Abstract has been revised and is now below 200 words.

  • 3 References should be checked:

         - The name of the microorganism needs to be written in italics (ref. 22, 23, 24, 25, 36, 37, 3941, 43, 46, 47, 49, 52, 53, 54).

         - The name of the journal should be abbreviated in all references.

         - Some references are uncompleted: pages number is missing in references 2, 9, 11, 28, 34, 36, 37, 40, 41, 49, 52, and volume in reference 11.

Thanks for pointing this out. Name of microorganism are now in italics in the reference part. The name of the journal has been abbreviated in all references. As a special note, please know that reference 2 has to be in this format, all the other references are now complete.

Reviewer 3 Report

Despite authors' aim is to provide a contribution to the exploitation of marine fungal diversity, their results basically consist in the finding of two known bioactive products, which rather could be indicative that the biotechnological potential of these fungi is low. Indeed, this makes this study neither innovative nor attractive to readers. Moreover, the preliminary approach based on searching genes involved in secondary metabolite production proved to be unfruitful and time/resource consuming. Indeed, in my opinion it is not at all surprising that the antimicrobial activity is unrelated to the presence of quite generic gene categories. Rather than representing an incongruent outcome, the statements at lines 148-150 and 159-162, coupled with the opposite case concerning P. olsonii, diminish the validity of the working hypothesis. However, in case the Editor does not consider these aspects prejudicial, I am favorable to the publication of this paper since it offers useful hints on structural variants and bioactivity of the two products. The text requires minor adjustments, as specified in the below list.

Title: correct to 'Oceanic'.

Line 38: ellipsis is not acceptable. Define what is missing.

Line 46: there is no reason to write 'Biosynthetic Gene Clusters' with uppercase initials. If later on using the abbreviation 'BGCs', define it here and use it in the rest of the manuscript.

Line 80: define the abbreviations 'NRPS', 'PKS' and 'TPS' on first mention.

Fig. 1 is difficult to read. Considering that phylogenetic relationships among strains have no relevance in this study, I suggest to report information contained in this figure in the form of a table, without distinguishing between yeasts and filamentous fungi.

End of line 152: 'or' to be written with uppercase initial.

Line 175: avoid repetition of the word 'antimicrobial' (e.g. replace 2nd one with 'antibiotic').

Line 205: first mention of Staphylococcus aureus is at line 157: use the full name there, and the abbreviation S. aureus in the rest of the text. This rule is also to be checked for all the other bacterial and fungal species names.

Line 215 and throughout the paper: separate quantities from measure units.

Line 216: it is not clear which 'other crude extracts' you are referring to.

Line 251: correct to 'produce'.

Reference citations must be checked in order to conform to the MDPI style.

Author Response

Despite authors' aim is to provide a contribution to the exploitation of marine fungal diversity, their results basically consist in the finding of two known bioactive products, which rather could be indicative that the biotechnological potential of these fungi is low. Indeed, this makes this study neither innovative nor attractive to readers. Moreover, the preliminary approach based on searching genes involved in secondary metabolite production proved to be unfruitful and time/resource consuming. Indeed, in my opinion it is not at all surprising that the antimicrobial activity is unrelated to the presence of quite generic gene categories. Rather than representing an incongruent outcome, the statements at lines 148-150 and 159-162, coupled with the opposite case concerning P. olsonii, diminish the validity of the working hypothesis. However, in case the Editor does not consider these aspects prejudicial, I am favorable to the publication of this paper since it offers useful hints on structural variants and bioactivity of the two products.

Thanks for these comments. We totally agree that no new marine fungal natural products were obtained here and that we rather provide information on the biotechnological potential of deep oceanic crust fungi through a bioprospecting-based approach. Actually, Marine Drugs offers the possibility to valorize this kind of data even if no novel structures were obtained, as for example the recently published papers on deep-sea fungi (Marchese et al. 2021), on actinomycetes (Handayani et al. 2021), sponge-associated bacteria (Gavriilidou et al. 2021) or on algae (Saraswati et al. 2021). We also agree on the fact that the genetic screening was not really successful here. However, we do think that our paper will give important information on this kind of genetic screening as we have, in our opinion, quite well discussed our data with several hypotheses to better understand these contrasted results. Our idea is that this kind of genetic screening can be viewed as a filter to lower the number of isolates to be screened but which is not exempt of biases. We have added a sentence to emphasize this statement in the text.

The text requires minor adjustments, as specified in the below list.

Title: correct to 'Oceanic'.

Thanks for pointing this out. This typo has been corrected

Line 38: ellipsis is not acceptable. Define what is missing.

Thanks for pointing this out. This typo has been corrected

Line 46: there is no reason to write 'Biosynthetic Gene Clusters' with uppercase initials. If later on using the abbreviation 'BGCs', define it here and use it in the rest of the manuscript.

Thanks for pointing this out. We have edited accordingly.

Line 80: define the abbreviations 'NRPS', 'PKS' and 'TPS' on first mention.

Thanks for this comment. We have defined the abbreviations on first mention.

Fig. 1 is difficult to read. Considering that phylogenetic relationships among strains have no relevance in this study, I suggest to report information contained in this figure in the form of a table, without distinguishing between yeasts and filamentous fungi.

Thanks a lot for this comment. Figure 1 has been improved. Figure 1 now contains the information as a dendrogram integrating the data related to biosynthetic gene clusters. We have decided to distinguish yeasts and filamentous fungi for 2 reasons: (i) this representation allows to easily compare the genetic potential between filamentous and yeasts, and (ii) dendrograms are based on genetic barcodes which are different (28S for yeasts and ITS for filamentous fungi), preventing from clustering all the isolates in one unique dendrogram or table.

End of line 152: 'or' to be written with uppercase initial.

Thanks for pointing this out. This typo has been corrected

Line 175: avoid repetition of the word 'antimicrobial' (e.g. replace 2nd one with 'antibiotic').

Thanks for pointing this out. We have edited accordingly.

Line 205: first mention of Staphylococcus aureus is at line 157: use the full name there, and the abbreviation S. aureus in the rest of the text. This rule is also to be checked for all the other bacterial and fungal species names.

Thanks for pointing this out. We have modified all bacterial and fungal names species as asked.

Line 215 and throughout the paper: separate quantities from measure units.

Thanks for pointing this out. We have edited accordingly.

Line 216: it is not clear which 'other crude extracts' you are referring to.

Thanks for pointing this out. To get the sentence clearer we modified as: ” . No activities were obtained from the crude extracts obtained from the other strains, …”

Line 251: correct to 'produce'.

Thanks for pointing this out. This typo has been corrected

Reference citations must be checked in order to conform to the MDPI style.

Citations have been checked to conform to MDPI style